# GOProFormer: A Multi-Modal Transformer Method for Gene Ontology Protein Function Prediction

**DOI:** 10.3390/biom12111709

**Published:** 2022-11-18

**Authors:** Anowarul Kabir, Amarda Shehu

**Affiliations:** 1Department of Computer Science, George Mason University, Fairfax, VA 22030, USA; 2Center for Advancing Human-Machine Partnerships, George Mason University, Fairfax, VA 22030, USA; 3Department of Bioengineering, George Mason University, Fairfax, VA 22030, USA; 4School of Systems Biology, George Mason University, Fairfax, VA 22030, USA

**Keywords:** multi-modal transformer, gene ontology, protein function

## Abstract

Protein Language Models (PLMs) are shown to be capable of learning sequence representations useful for various prediction tasks, from subcellular localization, evolutionary relationships, family membership, and more. They have yet to be demonstrated useful for protein function prediction. In particular, the problem of automatic annotation of proteins under the Gene Ontology (GO) framework remains open. This paper makes two key contributions. It debuts a novel method that leverages the transformer architecture in two ways. A sequence transformer encodes protein sequences in a task-agnostic feature space. A graph transformer learns a representation of GO terms while respecting their hierarchical relationships. The learned sequence and GO terms representations are combined and utilized for multi-label classification, with the labels corresponding to GO terms. The method is shown superior over recent representative GO prediction methods. The second major contribution in this paper is a deep investigation of different ways of constructing training and testing datasets. The paper shows that existing approaches under- or over-estimate the generalization power of a model. A novel approach is proposed to address these issues, resulting in a new benchmark dataset to rigorously evaluate and compare methods and advance the state-of-the-art.

## 1. Introduction

An explosion in the number of known protein sequences is now allowing us to leverage the Transformer [1] architecture to build Protein Language Models (PLMS) [2,3,4]. PLMs are highly appealing due to their ability to learn task-agnostic representations of proteins. In particular, they provide an alternative framework to link protein sequence to function without relying on sequence similarity. Sequence representations learned via PLMs have been shown useful for various prediction tasks, from predicting secondary structure [4], subcellular localization [4,5], evolutionary relationships within protein families [6], superfamily [7], and family [8] membership.

PLMs have yet to be demonstrated as useful for protein function prediction, which remains a hallmark problem in molecular biology [9]. In particular, throughput technologies have greatly increased the number of protein sequences in public repositories, but only about 1% of the sequences in the UniProtKB database have been functionally characterized [10]. This gap motivates computational approaches [11], and the computational literature on protein function prediction is rich [12].

In this paper we focus on challenging, community-driven instantiation of protein function prediction that utilizes the gene ontology (GO) hierarchy. The GO hierarchy consists of terms/concepts via which one can describe protein functions at varying resolution [10]. The GO framework is split into three sub-ontologies: the Cellular Component (CC), the Molecular Function (MF) and the Biological Process (BP). Each sub-ontology is organized as a directed acyclic graph (DAG) that encodes the relationships between the GO terms in a sub-ontology. The True Path rule is used [10] to associate proteins to GO terms. If a protein is annotated with a particular GO term *t*, it is also annotated with all the ancestor terms of *t* in the DAG of the sub-ontology to which *t* belongs. If a protein is not annotated with a particular GO term *t*, it is then also not annotated with any of the descendants of *t*.

GO annotation is a well-formulated instantiation of protein function prediction. It remains an open problem, though much progress has been made over the years, particularly due to deep models [13,14,15,16]. However, PLMs have yet to be demonstrated useful for GO annotation prediction. DeepChoi [17], a method presented in an article that remains in preprint, is the only occurrence of a PLM-based GO annotation method.

Currently, there is little to no understanding of how transformer-based approaches perform compared to the state-of-art for GO term prediction. This paper addresses this gap in the research literature. In particular, the paper makes two key contributions, one regarding methodology, and the other regarding rigorous training and testing data construction.

The paper debuts a novel method, GOProFormer, which leverages the Transformer [1] architecture in two ways. First, a sequence transformer encodes protein sequences in a task-agnostic feature space. Second, a graph transformer model learns a representation of the various GO terms that respects the hierarchical relationships among the terms. Additionally, a novel approach that treats a GO term as a concept to construct meaningful term representations is presented. The learned protein sequence and joint GO term representations are combined and utilized for multi-label classification, with the labels corresponding to GO terms. Analysis reveals a model that generalizes well. The model is shown superior over representative methods along CAFA3 evaluation metrics on increasingly challenging testing datasets.

The second major contribution in this paper is an investigation of various dataset construction protocols in related work. The paper shows that existing protocols may over- or underestimate the generalization power of a model. A new protocol is proposed here to address these issues and so rigorously compare methods. The paper makes another contribution; replacing the PLM in one of the baseline methods with a more powerful one results in an improved model.

The rest of this paper is organized as follows. Section 2 overviews related work. Details on the proposed method and data construction process are related in Section 3 and Section 4. The experimental evaluation is related in Section 5. The paper concludes in Section 6.

## 2. Related Work

Whether in protein structure prediction, protein function prediction, genome engineering, systems biology, or phylogenetic inference, deep neural networks are taking over as the state-of-the-art methods. Work in [18] provides a thorough review of deep learning literature in computational biology and shows the great diversity in neural network architectures for various domains. In particular, for protein function prediction one finds Convolutional Neural Networks (CNNs), Residual Networks (ResNets), Recurrent Neural Networks (RNNs), and Graph Neural Networks (GNNs).

Currently, PLMs are under-utilized for protein function prediction but increasing in momentum for other prediction tasks. For instance, work in [2] utilizes the popular ELMo language model to obtain vector representations of protein sequences. The representations are then shown effective for residue-level tasks, such as prediction of secondary structure and intrinsically-disordered regions, as well as protein level tasks, such as predicting subcellular localization and classifying whether a protein is water-soluble or membrane-bound.

Work in [8] debuts the PRoBERTa model. The model is pre-trained to learn task-agnostic sequence representations of amino-acid sequences. Since there is no inherent notion of words in a given amino-acid sequence, the authors in [8] restrict the vocabulary size to 10,000 words obtained via the byte-pair encoding (BPE) [19] algorithm. The PLM is then fine-tuned to solve two prediction tasks, protein family memberships and protein-protein interactions. Work in [7] does away with the notion of words and instead attends to each amino-acid, thus learning amino-acid level embeddings and sequence-level embeddings. The work is also one of the first to additionally attend to the three-dimensional structure of a protein and so learn joint sequence-structure representations.

ProtTrans [4] proposes two auto-regressive PLMs (based on Transformer-XL [20] and XLNet [21]) and four autoencoder PLMs (based on BERT [22], ALBERT [23], Electra [24], T5 [25]) for the same tasks as [2]. Work in [26] pretrains SeqVec [2] and ProtBert-BFD [4] and then transfers GO annotations based on protein proximity in the embedding space. The method does not directly utilize the learned representations for prediction but rather reformulates pairwise similarity.

Work in [5] applies the PLMs in [2,3,4,26] to localization prediction without multiple-sequence-alignments (MSAs). The authors apply a softmax weighted aggregation mechanism to compute the final embeddings of a protein sequence. Work in [6] demonstrates that PLM can predict the local evolution within protein families. The study shows that the model can capture evolutionary dynamics and timescales.

Early work in [26] shows that similarity-based transfer of GO annotations, typically performed in sequence space, improves in sequence representation space (with a reported Fmax of 39%, 53%, and 59% for BP, MF, and CC, respectively). The state-of-the-art for GO term prediction is currently represented by the DeepGO [13] and its variant DeepGOPlus [14]. DeepGO [13] incorporates a CNN to learn sequence-level embeddings and combines them with knowledge graph embeddings obtained from Protein-Protein Interaction (PPI) networks. DeepGOPlus [14] uses convolutional filters of different sizes with individual max-pooling to learn dense feature representations of protein sequences embedded via one-hot encoding. The authors show that combining the outputs from CNN with homology-based predictions improves accuracy and outperforms DeepGO.

Work in [15] proposes DeepGOA, a GCN-based model that additionally utilizes GO annotations and hierarchy to measure GO term correlations with which to update the edge weights of the DAG corresponding to a GO sub-ontology. The GCN is applied to the updated DAG to learn the semantic representation and latent inter-relations of the GO terms. Separately, a CNN learns the feature representation of the protein sequences with respect to the semantic representations. A dot product combines the two representations and allows training the whole network in an end-to-end fashion. Since no code is provided with the work in [15], we do not include it in list of methods for comparison. Later work in [16] (appearing earlier as DeepFUNC [27]), debuts a model also named DeepGOA, which extracts global sequence semantic representations via word2vec, local subsequence-based features extracted via InterPro to obtain motifs and domains, and combines the local and global features via a multi-scale CNN and a bi-directional long-short term memory (Bi-LSTM). The Deepwalk algorithm is additionally utilized over the protein-protein interaction (PPI) network to obtain a PPI-level embedding. The sequence and network are concatenated to predict protein functions.

Though only available in pre-print, DeepChoi [17] follows the overall approach in the 2019 DeepGOA [15] of learning sequence- and GO term representations and then combining them via a dot-product. The main difference is that the sequence representation is learned via SeqVec. We will refer to DeepChoi as DeepChoi-SeqVec from now on, to distinguish it from a new model we develop and report in here, where we replace SeqVec with a more powerful PLM.

A comparison of published literature indicates that DeepGOPlus outperforms DeepGO but performs similarly to DeepGOA, and DeepChoi-SeqVec outperforms DeepGOPlus, though the experimental evaluation is limited, and the testing dataset is rather small. Taken altogether, we consider DeepGOPlus and DeepChoi-SeqVec as representative baseline methods to which we compare GOProFormer.

## 3. Methodology

Let us assume that we have *N* proteins and *M* GO terms. Each protein is annotated with a non-empty subset of the GO terms. Treating GO terms as class labels casts protein function prediction as a multi-label classification problem. We organize the description of GOProFormer as follows. We first describe the input representation for proteins and GO terms. We then detail the process via which embeddings are obtained for proteins and GO terms. Finally, we describe how these embeddings are combined for multi-label classification. Figure 1 summarizes the overall model architecture.

### 3.1. From Input Representation to Learned Embedding of a Protein Molecule

As DeepGO [13], DeepGOPlus [14], DeepGOA [15], DeepChoi [17], and others, GOProFormer utilizes the amino-acid sequence of a protein and learns an embedding of a given protein sequence in a *D*-dimensional space. Specifically, considering protein 1≤i≤N in a given dataset, and its sequence $q_{i}$ of amino acids, GOProFormer learns the corresponding representation $s_{i}\in R^{q_{i}\times D}$; in this representation, each amino acid of the protein is mapped into $R^{D}$. This embedding is obtained via a language model.

#### 3.1.1. Amino-Acid Level Embeddings

We employ a state-of-the-art general purpose Transformer PLM, the Evolutionary Scale Modeling (ESM) [28] to extract amino-acid level features. ESM has been trained on 250 million protein sequences (a total of 86 billion amino acids) on masked-language-modelling tasks. There are several ESM-1 variants. For computational expediency, we utilize the lighter ESM-1 variant which has 12 layers and 85 M parameters (as opposed to, for instance, the ESM-1b variant with 33 layers and 650 M parameters).

#### 3.1.2. Protein Embedding

Given a learned $s_{i}\in R^{q_{i}\times D}$, GOProFormer obtains a protein-level representation $p_{i}\in R^{1\times D}$ by taking the average of the learned amino-acid-level features over the sequence length as in:(1)$$p_{i}=\frac{1}{q_{i}}\sum_{j=0}^{q_{i}}s_{ij}$$

The protein representation $p_{i}$ is then projected onto a d<D-dimensional space via a linear combination of the elements and a non-linear activation function as follows:(2)$$p_{i}^{{}^{\prime }}=ReLU\left(FCL\left(p_{i}\right)\right)$$

The obtained $p_{i}^{{}^{\prime }}$ is the learned embedding for a protein *i* in the dataset. Note that d<D, to discourage overfitting and capture meaningful information in a lower-dimensional space.

### 3.2. From Input Representation to Learned Embedding of a GO Term

In current literature, the input representation for GO terms is an one-hot encoding vector. For instance, even the most recent DeepGOA method [16] relies on such encodings and then uses a GCN to learn the joint representation of the GO terms while respecting the GO hierarchy. The problem with such an approach is that the nodes representation matrix is an identity matrix which does not contain any features that define the GO terms meaningfully. If we think of GO terms as concepts, a fundamental question is what does a concept mean? We answer that question by providing examples, and this motivates the novel approach below in GOProFormer.

Specifically, to represent the $j^{\mathrm{th}}$ GO term, where 1≤j≤M, we introduce a term pool $T_{j}$ that contains all the proteins in the training dataset that are annotated with that GO term. At each epoch of training, we sample once from all the pools (corresponding to the different GO terms) to generate the GO terms representation. We sample *K* prototype proteins from the respective pool, such that no prototypes are in the batch of this epoch. At validation and testing of the model, the same process is followed, as there are no overlaps among the proteins in the training, validation, and testing datasets.

Let us denote the embedding we want to learn from the $j^{\mathrm{th}}$ GO term as $t_{j}$. This embedding is learned in iterations. Let us assume that $t_{j}^{0}\in R^{K\times D}$ is the initial representation of the $j^{\mathrm{th}}$ GO term, defined as $t_{j}^{0}=\left[p_{0},\ldots ,p_{K-1}\right]$, where $p_{i}\in T_{j}$, $p_{i}\notin B$, and *B* denotes the set of proteins in the current batch.

GOProFormer uses the attention mechanism [29] to combine the features of the prototypes and compute the vector representation $t_{j}$ of a GO term. First, the method applies a non-linear activation function, tanh, on the features and then projects the activations onto one-dimensional space using a fully connected layer (FCL). Then it utilizes softmax to compute the attention-score, $\alpha_{j}$, for the *K* prototypes, as in:(3)$$\alpha_{j}=Softmax\left(FCL\left(tanh\left(t_{j}^{0}\right)\right)\right)$$

Finally, GOProFormer computes the weighted summation of the prototypes using the attention scores and applies an FCL to project the embedding onto a *d*-dimensional space as in Equation (Equation 4). Note that the FCLs in Equations (Equation 3) and (Equation 4) are independent of each-other.
(4)$$t_{j}^{1}=FCL\left(\sum_{k=0}^{K-1}\alpha_{j,k}p_{k}\right)$$

### 3.3. Transformer Encoder for GO Terms’ Embeddings

GOProFormer uses the vanilla Transformer encoder architecture [1] to learn the joint representation of the GO terms in a *d*-dimensional space. GOProFormer encodes the terms relation (DAG) as an adjacency matrix $\tilde{A}\in R^{M\times M}$, where ${\tilde{A}}_{i,j}$ is 1 if the $i^{\mathrm{th}}$ GO term has a parent or child relation with the $j^{\mathrm{th}}$ GO term, otherwise 0. Then, the self-loop information is added for each GO term to learn the self-representation as $A=\tilde{A}+I$, where *I* is the identity matrix.

GOProFormer applies *L* encoder layers with *H* attention heads at each layer. Computation in a head of $l^{\mathrm{th}}$ layer is carried over the following three steps. The attention score between two GO terms *i* and *j* is computed first, using their vector representation and relation between them.
(5)$$\begin{array}{c} \alpha_{i,j}=attention\left(t_{i}^{l},t_{j}^{l},A_{i,j}\right)\\ =Softmax[A_{i,j}\mp \left\{\frac{\left(t_{i}^{l}Q\right){\left(t_{j}^{l}K\right)}^{T}}{\sqrt{d}}\right\}] \end{array}$$

In the above, ∓ denotes the mask operation. This will cancel out the attention between *i* and *j* term if $A_{i,j}$ is zero, indicating no relation between them.

Second, GOProFormer computes an intermediate representation of the $i^{th}$ GO term using neighborhood aggregation:(6)$$z_{i}^{l}=\sum_{j=0}^{M-1}\alpha_{i,j}\left(t_{j}^{l}V\right)$$

In Equations (Equation 5) and (Equation 6), *Q*, *K* and *V* are learnable weight matrices. An FCL and residual connection are then applied to obtain the final embedding:(7)$$t_{i}^{l+1}=t_{i}^{l}+z_{i}^{l}+FCL\left(t_{i}^{l}+z_{i}^{l}\right)$$

At the end of the *L* encoder layers, GOProFormer has the jointly-learned representation of all the GO terms, denoted as $H={\left[t_{0}^{L},\ldots ,t_{M-1}^{L}\right]}^{T}\in R^{M\times d}$.

### 3.4. Prediction Layer

GOProFormer first applies the dot product of the vector representation of the $i^{\mathrm{th}}$ protein and $j^{\mathrm{th}}$ GO term and then applies the Sigmoid function to predict the final association score, $y_{i,j}^{\prime }$, as in:(8)$$y_{i,j}^{\prime }=Sigmoid\left(p_{i}^{\prime }\cdot t_{j}^{L}\right)$$

Equation (Equation 8) can be interpreted as follows; the model is $y_{i,j}^{\prime }$ percent confident (scaled into percentage) that the $j^{\mathrm{th}}$ GO term is associated with the $i^{\mathrm{th}}$ protein (or that the $i^{\mathrm{th}}$ protein is annotated with the $j^{\mathrm{th}}$ GO term). The dot product with the sigmoid is analogous to the normalized cosine similarity of two vectors in their shared vector space. This also justifies the joint representation learning of the GO terms from the pool of available proteins that we have described above. The final prediction is done by applying the true-path rule for a predicted confidence score by choosing a threshold.

### 3.5. Loss Function

GOProFormer utilizes the average binary-cross entropy loss to compute the multi-label term association loss for the $i^{\mathrm{th}}$ protein being annotated with the $j^{\mathrm{th}}$ term:(9)$$L=\frac{1}{N\times M}\sum_{i=0}^{N-1}\sum_{j=0}^{M-1}y_{i,j}log\left(y_{i,j}^{\prime }\right)+\left(1-y_{i,j}\right)log\left(1-y_{i,j}^{\prime }\right)$$

In Equation (Equation 9), $y_{i,j}$ denotes the ground-truth labels, where the *j*-th term is a positive example if $y_{i,j}$ is 1, and negative otherwise.

### 3.6. Implementation Details

As is practice, we train three different GOProFormer models for each sub-ontology on. Since PLMs introduce a large number of trainable parameters, for instance ESM1 [28] with 12 encoder layers has 85 M parameters, and the full *Saccharomyces cerevisiae* dataset we consider here contains only 6721 proteins, we freeze the PLM during the training and evaluation. One can pre-compute the proteins’ embeddings which will save time in training and evaluation. The embedding dimension (*D*) is the default 768 in ESM-1 and its variants. The hyperparameters related to learning GO terms representation are the number of prototypes sampled from the pool (*K*), the term embedding dimension (*d*), the number of encoder layers (*L*), and the number of attention heads in each encoder layer (*H*). These values are set as 5, 256, 3 and 2, respectively. The rationale behind choosing small values for *L* and *K* is to avoid a large number of parameters, so that the model will not overfit. The dimension of the feed-forward network after each encoder layer is set to 4 times the term embedding dimension *d*. The other hyperparameters related to model training, such as learning rate, mini-batch size and number of epochs, are set to 1 $\times \;10^{-4}$, 32 and 400 respectively. A dropout rate of 50% after each FCL except for the final prediction layer and L2-regularization with 0.01 at the loss function are applied to avoid the overfitting issue. Note that we do not use positional encoding of the GO terms, since they do not pose any inherent relative positioning as in a sequence.

## 4. Datasets

The question of how to construct the input dataset is central but under-investigated for its impact on the model and its performance in related literature. We set up the importance of answering this question briefly. All related work confirms that a model trained on the BP sub-ontology will perform worse than that model (trained separately) on the MF sub-ontology, which, in turn, will perform worse than that model (trained separately) on the CC sub-ontology. In other words, abusing notation, a partial ordering BP, MF, CC is established with regards to prediction task difficulty. The question of why this is the case has not been answered. This is one of the questions we answer in this paper.

Another question concerns the generalization of a model. This question is not addressed in related literature. Typically, the focus is on showing better performance on a few accepted metrics on the testing dataset, even if this dataset is demonstrably small compared to the training dataset. Details on how a model performs on the validation dataset are typically not related; neither can we find any details of analysis on the label distributions and potential differences among the training, validation, and testing dataset. We provide such insight here, and this insight motivates our proposal of a new, third dataset construction protocol.

We now describe and analyze in detail the two most prominent ones in literature and the one we propose. The analysis reports on the distribution of the number of labels (corresponding to GO terms) per protein in the training, validation, and testing datasets for each of the sub-ontologies (BP, CC, and MF).

The very first step shared by both protocols is to design the GO hierarchy data from The Gene Ontology Resource http://geneontology.org/docs/download-ontology/ [10,30] (filename: go.obo, releases/7 January 2022) which contains 30,497, 4471 and 12,488 terms involving three sub-ontologies, BP, CC and MF, respectively. One then downloads the GO annotations of Yeast from Gene Ontology Annotation (GOA) Database https://www.ebi.ac.uk/GOA/yeast_release (releases/27 July 2022) which contains 100,124 annotations for 6049 proteins. Last, manually-reviewed Yeast proteins are downloaded from UniProtKB/Swiss-Prot [10], a total of 6721 sequences. As is common practice, one develops three different models for each sub-ontology (BP, CC, and MF). The two training-validation-testing dataset construction protocols found in literature are the time-delay no-knowledge (TDNK) and the random-split (RS) ones. We propose a third, the time-series no-knowledge (TSNK) protocol. We describe each of these next and analyze them for what they reveal on the above questions.

### 4.1. Time-Delay No-Knowledge (TDNK)

This setting is adopted in CAFA (the Critical Assessment of Protein Function annotation) challenge. However, the distribution of the number of labels per protein may be different in the testing set than the training and validation sets in this setting. The steps of the TDNK protocol are as follows: (1) Annotations with non-experimental evidence codes are excluded. Experimental codes are EXP, IDA, IPI, IMP, IGI, IEP, TAS, IC, HTP, HDA, HMP, HGI or HEP. More information about GO evidence codes can be found in the GO evidence codes webpage at The Gene Ontology Resource http://geneontology.org/docs/guide-go-evidence-codes/ (releases/7 January 2022). (2) All annotations before a submission deadline T0 (11 August 2020) are used for the model development (dev-set). We keep this T0, so we can directly compare it to the published works. (3) No-knowledge proteins having at least one experimental annotation at T1 (14 January 2022) are used for mode testing. Note, no-knowledge indicates that the test-set proteins do not have overlapping proteins with the dev-set (this includes the training and validation datasets). (4) The training (train-set) and validation (val-set) sets are obtained by respectively sampling 80% and 20% of the data at random from the dev-set. (5) Apply the true path rule to expand annotations. Specifically, if a protein is annotated with a GO term, then the same protein is also annotated with all the ancestors of that GO term. (6) Rank GO terms by their number of associations and select terms with the minimum number of 150, 25 and 25 for BP, CC, and MF, respectively. We refer to these GO terms as study-terms. (7) Exclude GO terms other than study-terms. (8) Exclude proteins annotated with no study-terms. (9) Exclude proteins that are not in the UniProtKB/Swiss-Prot [31] database.

Table 1 shows the distribution of labels (GO terms) for BP, CC, and MF in the TDNK train-, val-, and test-set. A distribution is summarized with the mean and standard deviation. Figure 2 relates the distributions. Table 1 and Figure 2 expose the rationale why related literature reports that BP is a more difficult prediction task, followed by MF and CC. It is clear that the mean number of labels in the test-set is much lower than in the train- and val-set for BP; the means in the train- and val-sets are comparable. In contrast, for MF and CC the mean number of labels for the train-, val- and test-set are comparable. This explains the higher difficulty with predicting in the BP sub-ontology. It does not inform, however, on the reported higher difficulty in MF over CC; label imbalance (note the rather large standard deviations) may additionally explain this phenomenon observed for related works (DeepGO, DeepGOPlus, and others); our results on TDNK in Section 5 do not show that MF is harder than CC for GOProFormer.

#### 4.1.1. Random-Split (RS)

We also evaluate using random-split val- and test-set as in DeepGO [13]. All annotations with experimental evidence codes at T1 (14 January 2022) are considered as the dev-set. The test-set consists of 15% data points sampled at random from the dev-set. The train-set and val-set are composed by taking 85% and 15% of the data points at random from the remaining dataset. All the other steps, such as exclusion and applying true-path rule, are applied. Although the train-, val- and test-set do not share any proteins among them, the distribution of labels per protein is similar due to uniform sampling. Table 1 indeed shows this to be the case. The mean number of labels is now similar for all three, the train-, val-, and test-set for BP, as well.

#### 4.1.2. Time-Series No-Knowledge (TSNK)

We propose a new protocol, inspired by the need to improve on label distribution. Our insight is that the full dataset can be treated as time-series data. Specifically, the timeline from T0 (1 January 2000) to T1 (14 January 2022) is considered as a time-series which is then consecutively divided into non-overlapping timelines. Then, 20% of the timelines are considered as test-timelines. From the remaining, 10% and 90% timelines are considered as val- and train-timelines. The experimental annotations published in the defined timelines create the train-, val- and test-set. Any annotations from the val- and test-set whose protein is in the train-set are removed from the val- and test-set and only kept into the train-set. Thus, there is no-knowledge between the train and val/test-set. We note that all the other steps, such as exclusion and applying true-path rule are applied.

Table 1 shows the utility of this approach. Unlike TDNK and as in RS, the label distributions between the val-set and test-set are more similar (on each sub-ontology). The standard deviations in each sub-ontology are smaller than in TDNK and RS on the val- and test-set. TSNK promises to lead a model to better generalizability than TDNK and RS.

It is worth noting that TDNK may generate a very small number of proteins and their associated annotations, which may not follow the distribution of the training and validation sets, as shown Figure 2 test-set. Thus, the actual generalization power of a model may be underestimated. On the other hand, RS does not guarantee the no-knowledge test-set. Thus, the model may suffer from data leakage, and in turn the performance of a model may be overestimated.

## 5. Result

### 5.1. Experimental Setup

For the purpose of experimental evaluation, we follow the CAFA suggested evaluation criteria, Fmax and Smin [32,33]. Fmax is the maximum harmonic mean of precision and recall across all possible thresholds on the predicted protein–Go-term association matrix. Specifically,
(10)$$Fmax=\max_{th}\frac{2\cdot prec(th)\cdot rec(th)}{prec(th)+rec(th)}$$
where prec(th) and rec(th) denote the average precision and recall score at threshold th∈[0,1]. The higher the Fmax obtained by a model, the better the performance.

Smin computes the semantic distance between predicted and ground-truth annotations based on average remaining uncertainty (ru(th)) and misinformation measure (mi(th)) over a decision boundary th∈[0,1]. Specifically,
(11)$$Smin=\min_{th}\sqrt{ru{\left(th\right)}^{2}+mi{\left(th\right)}^{2}}$$
where
(12)$$ru\left(th\right)=\underset{i}{\mathrm{avg}}\sum_{a\in (y_{i}-y_{i}^{\prime })}IC\left(a\right)$$
and
(13)$$mi\left(th\right)=\underset{i}{\mathrm{avg}}\sum_{a\in (y_{i}^{\prime }-y_{i})}IC\left(a\right)$$

In the above, $y_{i}$ and $y_{i}^{\prime }$ denote the ground-truth and predicted annotations for $i^{\mathrm{th}}$ protein and IC(a) defines the log-scaled class prior as information-content of annotation *a* as IC(a)=−log(a|Parent(a)). The lower the Smin obtained by a model, the better the performance.

In addition to these two protein-centric evaluation measures, we utilize a GO term-centric measure that is popular for multi-class classification, the area-under-precision-recall curve (AUPRC). We choose AUPRC over AUC, as AUPRC is more sensitive to class-imbalance than AUC.

As related in Section 4, the evaluation is carried out over three separate settings, TDNK, RS, and TSNK.

We first provide a detailed look into the GOProFormer performance and then compare it to DeepGoPlus [14], DeepChoi-SeqVec [17], and DeepChoi-ESM-1, where we replace the SeqVec language model with the more powerful ESM-1. Support for the AllenNLP library employed by DeepChoi for SeqVec has been discontinued, so we present a new variant of DeepChoi, DeepChoi-ESM-1 that utilizes the ESM-1 PLM also employed in our proposed GOProFormer.

### 5.2. Model Performance on Train-, Val-, and Test-Set

We report the training and validation binary cross-entropy loss, as well as the validation Fmax performance of GOProFormer on each of the dataset settings versus (vs.) the number of epochs in Figure 3, Figure 4 and Figure 5. To avoid overfitting, we save the best model at best performance on the validation set while training.

### 5.3. Comparative Performance on TDNK Datset

Table 2 shows the performance of DeepGoPlus, DeepChoi-ESM-1, and GOProFormer on the TDNK val- and test-set over each of the GO sub-ontologies and for each of the evaluation metrics. For completeness, we also add the results reported in [26], though the work transfers annotations based on similarity over embeddings. The highest values in each category are highlighted in boldface font. If we rely on reported values on the TDNK dataset, DeepChoi-SeqVec reports test-set Fmax values of 0.518, 0.470, 0.637 and AURPC values of 0.476, 0.368, and 0.626, respectively, on the MF, BP, and CC sub-ontologies in the original paper [17].

Several observations can be drawn from Table 2. The difference in performance between the test-set (lower) and the val-set shows that all methods’ generalization potential is underestimated. This further confirms the distribution difference argument between val- and test-set we outline in Section 4. The performance of all methods follows the known trend of higher difficulty on BP than MP and CC. GOProFormer and DeepChoi-ESM-1 narrow the difference in performance between BP, CC, and MF over the other methods. DeepChoi claims the best values on the BP and MF on the test-set, followed closely by GOProFormer (DeepChoi-ESM-1 outperforms DeepChoi-SeqVec on the test set).

### 5.4. Comparative Performance on RS Dataset

Table 3 shows the performance of DeepGoPlus, DeepChoi-ESM-1, and GOProFormer on the RS val- and test-set over each of the GO sub-ontologies and each of the evaluation metrics. As before, the highest values in each category are highlighted in boldface font. GOProFormer outperforms all methods on the val- and test-set with two exceptions. This is due to the more comparable class distribution between these sets (see Table 1 in Section 4). Differences in performance between the test-set (lower) and the val-set are reduced for all models, removing concerns of overfitting. BP prediction remains more difficult.

### 5.5. Comparative Performance on TSNK Dataset

Table 4 shows the performance of DeepGoPlus, DeepChoi-ESM-1, and GOProFormer on the TSNK val- and test-set over each of the GO sub-ontologies and each of the evaluation metrics. The highest values in each category are highlighted in boldface font. With one exception, GOProFormer obtains the highest performance on each metric, for each of the GO sub-ontologies, over both the val- and test-set, over all methods. Differences in performance between the test-set (lower) and the val-set are low, alleviating concerns of overfitting. Again, BP prediction remains more difficult.

### 5.6. Impact of Datasets on Performance: TDNK vs. RS vs. TSNK

The above analysis focuses primarily on comparing methods on three separate datasets. We now expand further on the performance of GOProFormer and the TDNK, RS, and TSNK datasets to better understand the impact of a dataset distribution on performance, as shown in Table 1 and Figure 2. Comparing the performance of GOProFormer over the test-set in the TDNK and RS setting (see Table 2 and Table 3) reveals that the model achieves its best performance on all metrics (Fmax, Smin and AUPRC) in the RS setting (vs. TDNK). For instance, GOProFormer achieves an Fmax value of 0.577 vs. 0.389 (BP), 0.751 vs. 0.537 (CC) and 0.614 vs. 0.528 (MF) in the RS versus the TDNK setting, respectively. This in agreement with our observations in Section 4, as the distribution of labels per protein in the RS setting is similar between the train- and test-sets due to uniform sampling; in contrast, in the TDNK setting, there is a stark difference between the distributions of the train and test-set. Comparing the performance of GOProFormer over the test-set in the RS vs. the TSNK setting (see Table 3 and Table 4) reveals no clear differences according to the Fmax and AUPRC metrics. Differences in Smin are prominent. In the RS setting, GOProFormer achieves an Smin value of 15.657 (BP), 8.565 (CC), and 7.972 (MF). In the TSNK setting, GOProFormer achieves better Smin values of 8.810 (BP), 5.073 (CC), and 6.308 (MF). Smin is a more stringent measure of performance [16], and these results indicate that TSNK is a trade-off between the TDNK and RS settings and affords GOProFormer better generalization in comparison with other state-of-the-art models.

## 6. Conclusions

This paper has presented GOProFormer, a GO protein function prediction method that accounts for both protein sequence and the GO hierarchy in its learned representations. The method utilizes the transformer architecture; a language model to embed protein sequences in a semantic space, and a novel graph transformer to embed GO terms in a space that respects the GO terms relationships. The notion of a pool is introduced to learn more meaningful representations for GO terms.

While much of the computational literature on GO term prediction has acknowledged the importance of integrating the knowledge from GO annotations in a model, our work here is the first to show how one distill such knowledge and formulate it into a prior in a transformer-based model. It is worth noting that the GO hierarchy of terms via which one can describe protein functions at varying resolution represents an interesting source of biological priors which we integrate in a transformer-based model for protein prediction tasks situated in biological/domain knowledge.

Our method advances the state of the art. Moreover, another contribution of this work is that various datasets are carefully characterized and employed to evaluate performance and so understand the intrinsic characteristics of the different datasets utilized in related literature. Moreover, a novel protocol is described for training and testing dataset construction that avoids data leakage and facilitates model generalization. The comparative performance analysis on the three dataset settings shows that a models’ generalizability may be over- or under-estimated depending on the dataset.

PLMs are becoming increasingly important to developers and practitioners alike, and in this paper we extend their reach to a community-set challenge of GO term prediction. Many avenues remain open for future research. For instance, investigating the interplay between *D* and *d* and their impact on model performance is an important direction. Future research is additionally needed to understand more deeply what language models are capturing about the biological data and how this informs us on their power and their shortcomings. With so many frontiers open, this is an exciting time for computational biology researchers, and we expect PLMs to become increasingly appealing for various molecular biology prediction tasks.

## Figures and Tables

**Figure 1 biomolecules-12-01709-f001:**
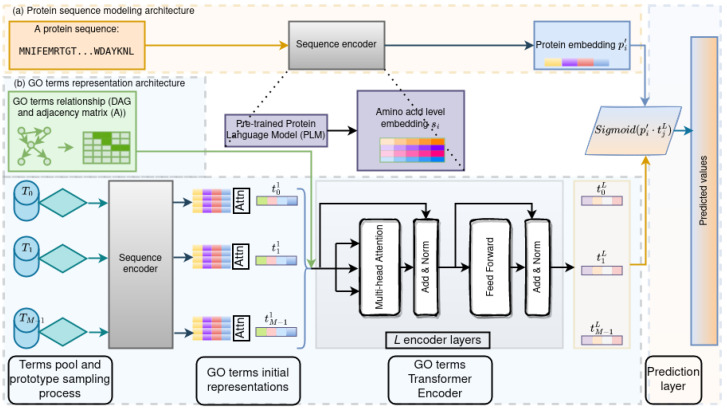
The figure shows the proposed multi-modal Transformer model architecture. (**a**) depicts a black-box protein sequence modeling encoder. (**b**) shows the GO terms representation module using the Transformer encoder architecture.

**Figure 2 biomolecules-12-01709-f002:**
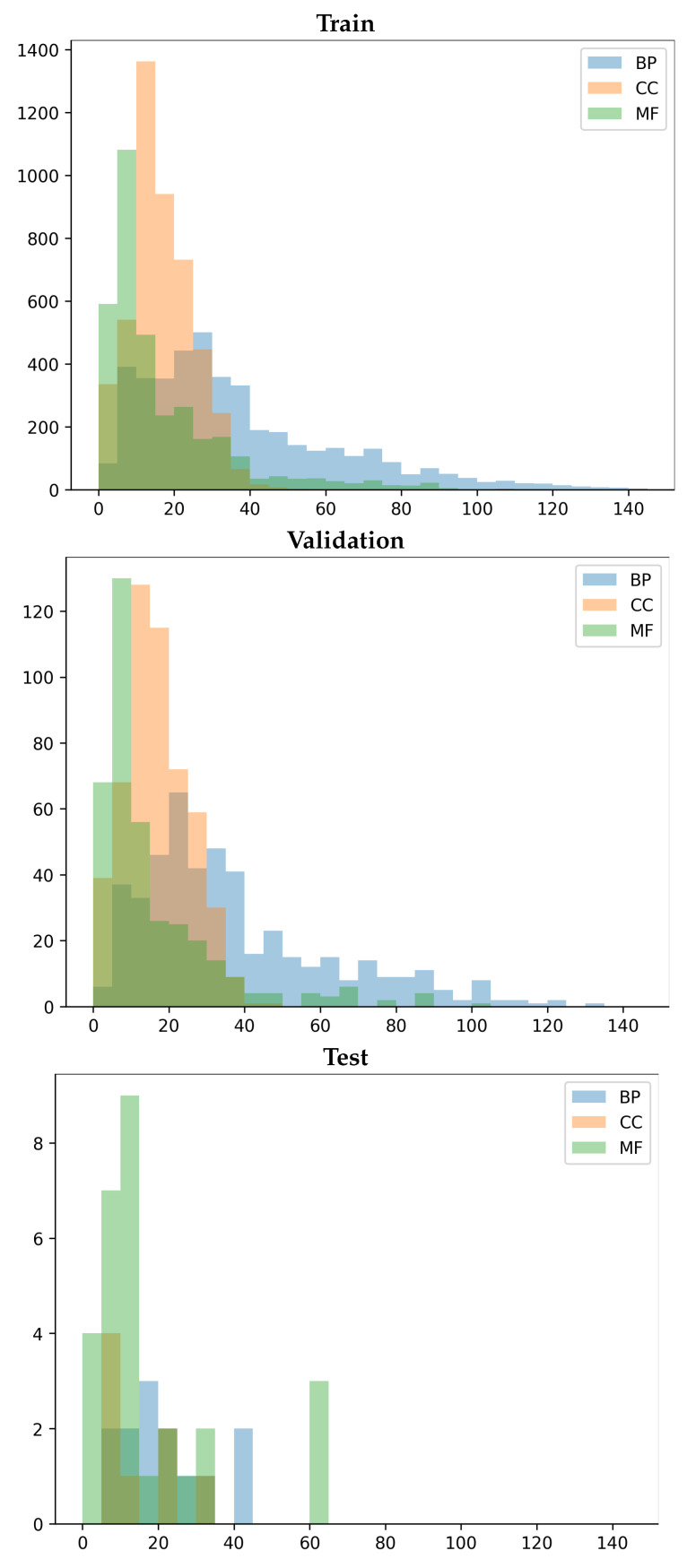
The figures show the distribution of the number of labels per protein for the BP, CC and MF classes in the train-, val-, and test-set, respectively, considering the cutoff values of 150, 25, and 25. The test-set is generated considering the TDNK split, and the val-set is generated by randomly sampling 10% of the full dataset.

**Figure 3 biomolecules-12-01709-f003:**
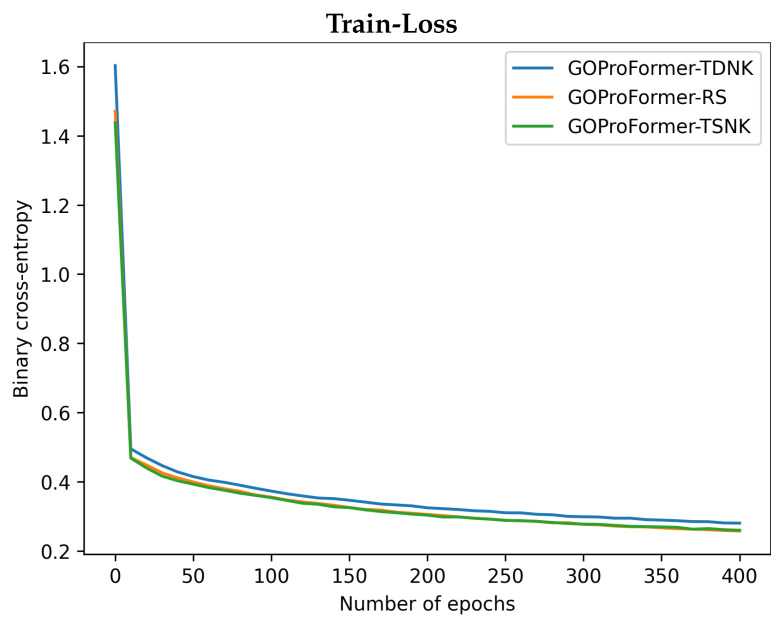
The train binary cross-entropy loss, validation binary cross-entropy loss and validation Fmax are shown for BP on each of the three dataset settings.

**Figure 4 biomolecules-12-01709-f004:**
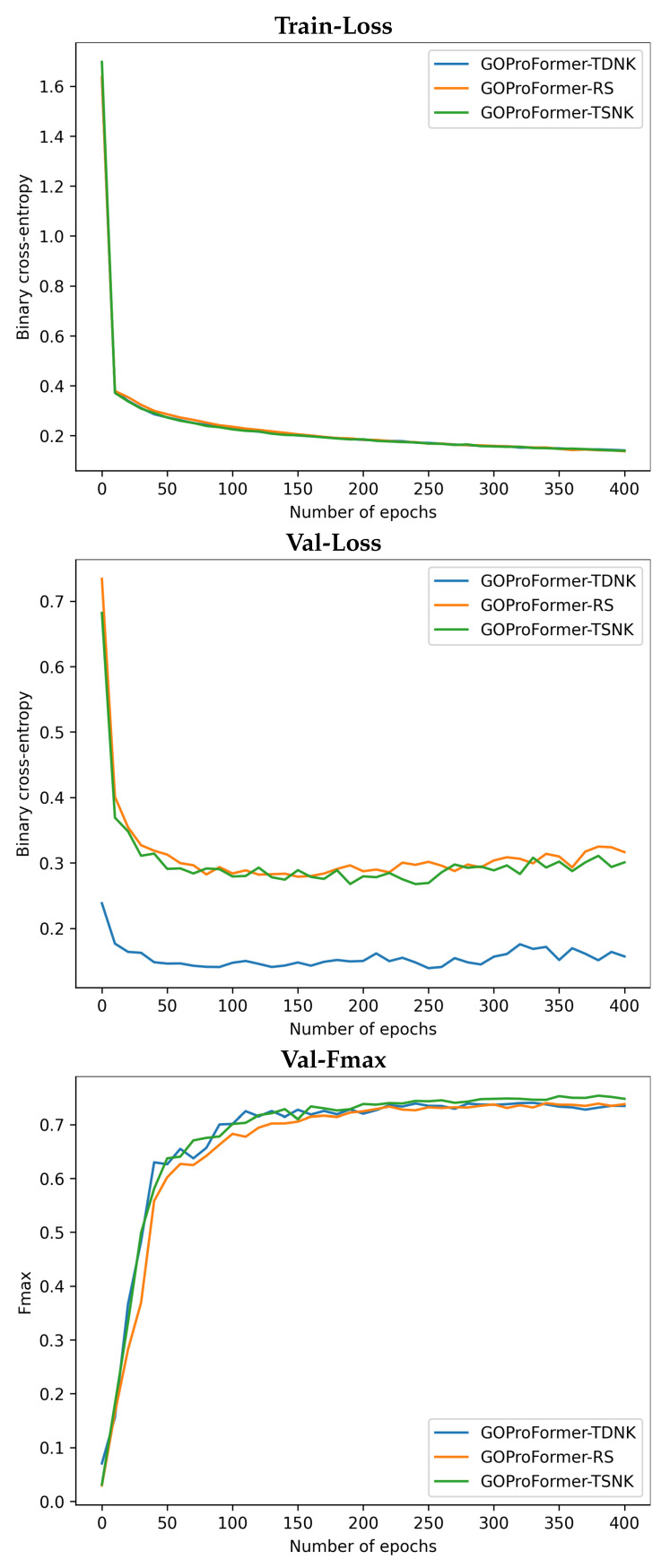
The train binary cross-entropy loss, validation binary cross-entropy loss and validation Fmax are shown for CC on each of the three dataset settings.

**Figure 5 biomolecules-12-01709-f005:**
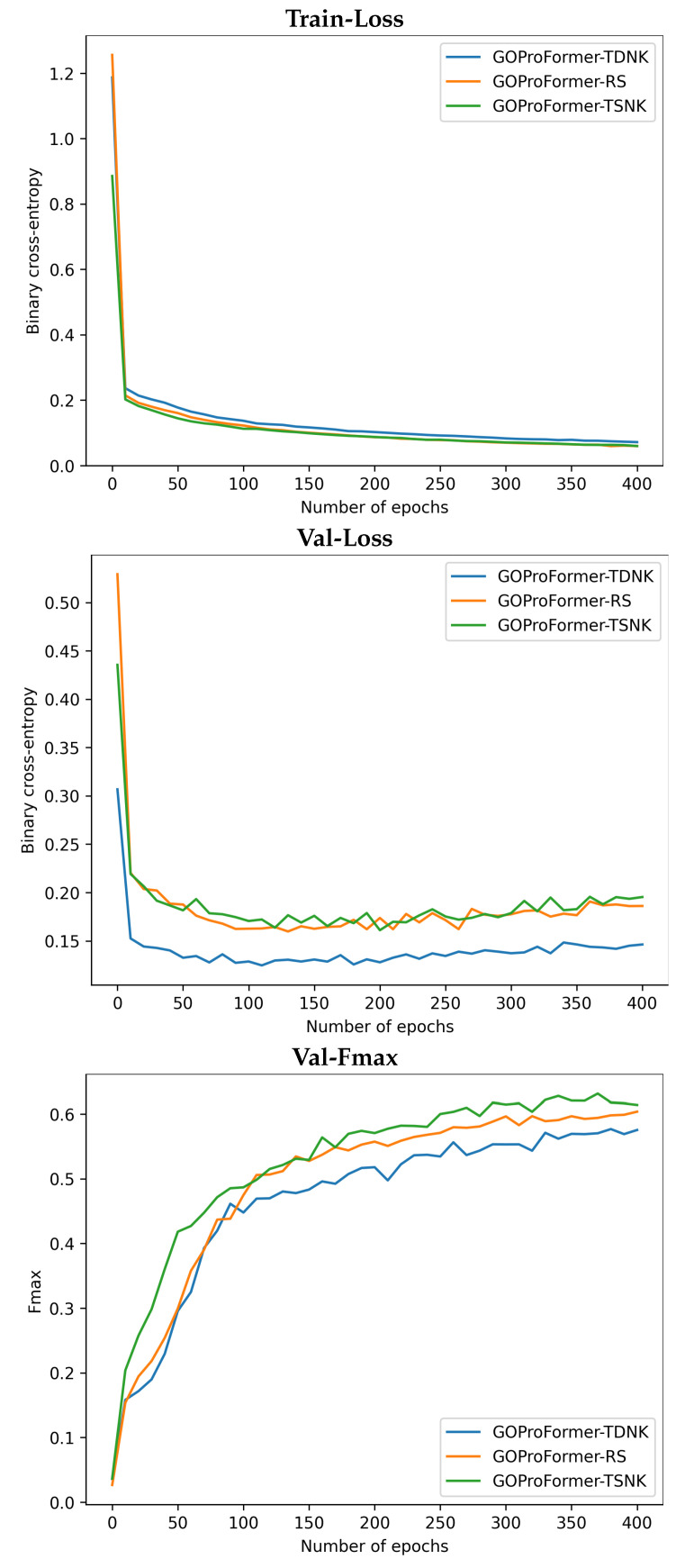
The train binary cross-entropy loss, validation binary cross-entropy loss and validation Fmax are shown for MF on each of the three dataset settings.

**Table 1 biomolecules-12-01709-t001:** The distribution of the number of labels (GO terms) per protein in the train-, val-, and test-set for each sub-ontology in the respective dataset generation processes (TDNK, RS, and TSNK). The distribution is summarized with the mean and standard deviation.

Dataset	GO	Train	Val	Test
TDNK	BP	36.198, 26.175	34.646, 24.914	21.600, 12.580
	CC	16.354, 8.249	16.523, 8.259	14.000, 11.019
	MF	17.085, 18.523	16.751, 17.176	17.448, 18.001
RS	BP	35.483, 25.608	35.512, 26.121	35.053, 25.726
	CC	16.435, 8.167	16.045, 8.492	16.311, 8.464
	MF	16.971, 18.232	17.738, 19.465	16.923, 18.190
TSNK	BP	37.472, 26.262	20.289, 17.017	19.471, 15.899
	CC	16.787, 8.315	11.626, 5.722	11.018, 6.204
	MF	17.762, 19.203	10.771, 11.877	11.603, 12.928

**Table 2 biomolecules-12-01709-t002:** Comparison of methods on TDNK dataset. Best values are highlighted in boldface font.

Metrics	GO	[26] Test	DeepGOPlus		DeepChoi-ESM-1		GOProFormer	
			Val	Test	Val	Test	Val	Test
Fmax	BP	0.39	0.482	0.337	0.556	**0.428**	**0.591**	0.389
	CC	0.59	0.721	0.504	0.738	0.510	**0.750**	**0.537**
	MF	0.53	0.403	0.461	0.569	**0.535**	**0.624**	0.528
Smin	BP	–	17.997	12.145	16.478	**11.697**	**15.259**	12.681
	CC	–	9.518	10.449	8.967	10.378	**8.611**	**9.588**
	MF	–	10.629	8.918	9.610	**8.478**	**8.349**	8.827
AUPRC	BP	–	0.486	0.295	0.590	**0.395**	**0.623**	0.338
	CC	–	0.679	0.372	**0.783**	0.483	0.782	**0.506**
	MF	–	0.353	0.362	0.590	**0.526**	**0.643**	0.468

**Table 3 biomolecules-12-01709-t003:** Comparison of methods on RS dataset. Best values are highlighted in boldface font.

Metrics	GO	DeepGOPlus		DeepChoi-ESM-1		GOProFormer	
		Val	Test	Val	Test	Val	Test
Fmax	BP	0.486	0.477	0.564	0.544	**0.589**	**0.577**
	CC	0.709	0.720	**0.736**	0.740	**0.736**	**0.751**
	MF	0.393	0.391	0.545	0.541	**0.607**	**0.614**
Smin	BP	18.525	18.621	16.436	16.856	**15.471**	**15.657**
	CC	10.273	10.570	8.783	9.233	**8.413**	**8.565**
	MF	10.570	11.269	10.027	9.588	**8.698**	**7.972**
AUPRC	BP	0.488	0.483	0.585	0.572	**0.629**	**0.627**
	CC	0.654	0.668	**0.773**	**0.790**	0.751	0.765
	MF	0.339	0.332	0.563	0.555	**0.617**	**0.619**

**Table 4 biomolecules-12-01709-t004:** Comparison of methods on TSNK dataset. Best values are highlighted in boldface font.

Metrics	GO	DeepGoPlus		DeepChoi-ESM-1		GOProFormer	
		Val	Test	Val	Test	Val	Test
Fmax	BP	0.460	0.491	0.499	0.529	**0.526**	**0.557**
	CC	**0.739**	0.709	0.738	0.712	**0.739**	**0.729**
	MF	0.457	0.436	0.524	0.541	**0.580**	**0.623**
Smin	BP	10.499	9.721	10.074	9.080	**9.600**	**8.810**
	CC	4.598	5.487	4.608	5.359	**4.541**	**5.073**
	MF	7.719	8.267	7.565	7.748	**6.338**	**6.308**
AUPRC	BP	0.439	0.469	0.466	0.495	**0.526**	**0.557**
	CC	0.691	0.649	**0.782**	**0.751**	0.724	0.693
	MF	0.368	0.350	0.513	0.517	**0.564**	**0.584**

## Data Availability

The datasets are available at http://geneontology.org/docs/download-ontology/ (releases/7 January 2022) and https://www.ebi.ac.uk/GOA/yeast_release (releases/27 July 2022).

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
