# Peer review of "GOProFormer: A Multi-Modal Transformer Method for Gene Ontology Protein Function Prediction"

_biomolecules, 2022, doi:10.3390/biom12111709_

Round 1

Reviewer 1 Report

This manuscript “GOProFormer: A Multi-modal Transformer Method for Gene Ontology Protein Function Prediction” submitted by Anowarul Kabir and Amarda Shehu reports a novel approach for GO protein function prediction. In the manuscript, the authors describe their approach and perform a comprehensive and comparative investigation of various relevant datasets. In my opinion the approach described is very interesting and the manuscript is of high quality.

Minor suggestion:

I) I would be interesting if the authors can provide a schematic graph showing the transformer/network architecture of the GOProFormer approach.

Author Response

GOProFormer: A Multi-modal Transformer Method for Gene Ontology 

Thank you for considering our manuscript and for giving us an opportunity to incorporate the reviewer feedback. We have now done so. All changes to the manuscript are in blue font. In addition to responding to each reviewer on each point raised, we have also conducted a close reading of our manuscript and have addressed any found typos or gramatical mistakes. 

Reviewer 1: 

Comment: It would be interesting if the authors could provide a schematic graph showing the transformer/network architecture of the GOProFormer approach. 

  • We have included a schematic graph in Figure 1 that shows the workflow and overall network architecture of the GOProFormer. We thank the reviewer for incentivizing us to put this together. 

Reviewer 2: 

Comment: Figure 1 and table 1-Please provide a description of the main results for both figure and table. 

  • We thank the reviewer for prompting us to this. We have now provided an expanded description in subsection 5.6 titled as “Impact of Datasets on Performance: TDNK vs. RS vs. TSNK”. 

Comment: Some paragraphs in conclusion fit better in the discussion section. I suggested to the authors to add discussion making comparison between the results obtained in the present work with previously described language model, emphasizing the novelty of the work. 

  • We thank the reviewer for these suggestions. We have expanded the discussion in the context of our Results section (e.g., in subsection 5.6) to reduce redundancies. We have expanded the Conclusion section to emphasize the novelty of the work, its state of the art performance, and the broader contributions of our work. 

Reviewer 2 Report

The present manuscript “GOProFormer: A Multi-modal Transformer Method for Gene Ontology Protein Function Prediction” presented a new method aiming protein function prediction combining learned sequence and GO terms. In fact, it might be a great contribution to researchers which work in protein function prediction. The manuscript is well-written, methodology is well-described, mathematical models and statics analysis were properly used to test authors’ hypothesis.

General comments

Results. Figure 1 and table 1-Please provide a description of the main results for both figure and table.

Discussion-Some paragraphs in conclusion fits better in discussion section. I suggested to the authors to add discussion making comparison between the results obtained in the present work with previously described language model, emphasizing the novelty of the work.

Conclusion-Please review as mentioned above.

Author Response

(The authors gave the same response as above.)
